# Are There Ethnic Differences in Recorded Features among Patients Subsequently Diagnosed with Cancer? An English Longitudinal Data-Linked Study

**DOI:** 10.3390/cancers15123100

**Published:** 2023-06-07

**Authors:** Tanimola Martins, Obioha C. Ukoumunne, Georgios Lyratzopoulos, Willie Hamilton, Gary Abel

**Affiliations:** 1Department of Health and Community Sciences, Faculty of Health and Life Sciences, College of Medicine and Health, University of Exeter St Luke’s Campus, Magdalen Road, Exeter EX1 2LU, UK; w.hamilton@exeter.ac.uk (W.H.); g.a.abel@exeter.ac.uk (G.A.); 2National Institute for Health and Care Research (NIHR) Applied Research Collaboration (ARC) South West Peninsula (PenARC), Department of Health and Community Sciences, Faculty of Health and Life Sciences, University of Exeter, Exeter EX1 2LU, UK; o.c.ukoumunne@exeter.ac.uk; 3Epidemiology of Cancer Healthcare & Outcomes (ECHO) Group, University College London, 1-19 Torrington Place, London WC1E 7HB, UK; y.lyratzopoulos@ucl.ac.uk

**Keywords:** ethnic inequalities, cancer symptoms, cancer diagnosis, primary care, diagnostic pathway, symptomatic cancer

## Abstract

**Simple Summary:**

This UK population-based study aimed to determine whether the presenting features of cancer recorded in primary care before diagnosis differed by ethnicity. We found that for some cancer types, Asian and Black patients were more likely than White patients to have ‘less concerning’ features, such as cough and upper abdominal pain, recorded before diagnosis. Indeed, there was no site where either group was more likely than the White group to have alarm features, such as blood in urine, recorded. However, further research is necessary to determine the extent to which these ethnic differences reflect the disease biology, patient, or healthcare factors.

**Abstract:**

We investigated ethnic differences in the presenting features recorded in primary care before cancer diagnosis. Methods: English population-based cancer-registry-linked primary care data were analysed. We identified the coded features of six cancers (breast, lung, prostate, colorectal, oesophagogastric, and myeloma) in the year pre-diagnosis. Logistic regression models investigated ethnic differences in first-incident cancer features, adjusted for age, sex, smoking status, deprivation, and comorbidity. Results: Of 130,944 patients, 92% were White. In total, 188,487 incident features were recorded in the year pre-diagnosis, with 48% (89,531) as sole features. Compared with White patients, Asian and Black patients with breast, colorectal, and prostate cancer were more likely than White patients to have multiple features; the opposite was seen for the Black and Other ethnic groups with lung or prostate cancer. The proportion with relevant recorded features was broadly similar by ethnicity, with notable cancer-specific exceptions. Asian and Black patients were more likely to have low-risk features (e.g., cough, upper abdominal pain) recorded. Non-White patients were less likely to have alarm features. Conclusion: The degree to which these differences reflect disease, patient or healthcare factors is unclear. Further research examining the predictive value of cancer features in ethnic minority groups and their association with cancer outcomes is needed.

## 1. Introduction

Identifying cancer among symptomatic patients is often complicated by the nature of symptoms reported during primary-care consultations [1,2]. In particular, most cancers present with low-risk (non-alarm) symptoms, such as cough and abdominal pain, which are also common in benign diseases and have low positive predictive values (PPVs) for cancer [3,4]. Alarm symptoms, such as haemoptysis, have higher PPVs, but are rarer presentations of cancer [5,6]. On the other hand, patients presenting with a combination of synchronous symptoms are more likely to raise a general practitioner’s (GP’s) suspicion compared to those with an isolated symptom [7,8]. In practice, however, GPs use their intuition alongside clinical guidelines and decision-support tools to determine whether a symptom (or combination of symptoms) warrants specialist investigation [1,2]. This process also relies on patients’ willingness and ability to articulate their symptomatic experiences during primary-care consultations. A recent multi-method study of UK men presenting with possible prostate cancer symptoms in primary care showed that men, particularly Black men, may not fully disclose their concerns or symptoms during initial consultations, partly due to the relatively short duration of primary-care consultations [9]. Non-disclosure of symptoms may explain the greater frequency of primary-care consultations [10] and the longer time to diagnosis among the UK Asian and Black groups compared with the White group [11], although this aspect has not been fully examined. In the present study, we used cancer-registry-linked data to investigate possible ethnic differences in the number and type of cancer features recorded in primary care before a cancer diagnosis. We hypothesised that the relevant recorded features of cancer in the year before diagnosis are similar across ethnic groups.

## 2. Materials and Methods

Study design and data sources: We performed a population-based study of English patients diagnosed with one of six common cancers using data from the Clinical Practice Research Datalink (CPRD-Aurum) with linkage to Hospital Episode Statistics (HES), and the National Cancer Registration and Analysis Service (NCRAS) cancer registry data. The scope and strengths of the CPRD-linked data are well documented [12,13]. The CPRD-Aurum (August 2019 release), comprised routinely gathered data from 890 consenting English practices, with over 28 million patients eligible for linkage to other health care databases [12,13]. It included coded and anonymised data on patients’ medical histories, including symptoms, investigations, diagnoses, prescriptions, referral, and demographics (e.g., age, gender, and ethnicity) [12,13]. HES data—Admitted patient care and Outpatient elective—contains medical records on all hospital admissions and outpatient appointments in England [12,13,14]. The NCRAS cancer-registry data includes records of all tumours diagnosed in England, alongside information about treatment and patient-reported outcomes [15].

Participants: Eligible participants were aged at least 40 years on the date of cancer diagnosis, with an incident of cancer recorded in the cancer registry between 1 January 2006 and 31 December 2016. We excluded patients diagnosed with a cancer unusual for their sex (e.g., female/prostate), and those diagnosed via screening or death certificate only. Furthermore, patients with no primary-care attendance, or with no cancer-specific features recorded in the year before diagnosis, and those with missing ethnicity records in the CPRD and HES (see below) were excluded. Differences between those with and without recorded features were explored (see Appendix A).

### 2.1. Study Variables

Cancer sites: Using the NCRAS data, we extracted patient records on the four most common cancers [lung (ICD10 C34), breast (C50), prostate (C61), colorectal (C18-C20)], and three cancers more commonly diagnosed in ethnic minority groups [oesophagus (C15), stomach (C16)), and myeloma (C90)] [16,17,18,19]. We merged oesophagus and stomach cancers into oesophagogastric cancer because they share diagnostic features and suspected-cancer referral criteria.

Ethnicity: Patients’ ethnicities were identified from CPRD codes, or HES data if missing in the CPRD, as recommended previously [20,21,22]. The processes involved in ethnicity data extraction are detailed elsewhere [21,22]. Briefly, we extracted and collapsed all ethnicity records from the CPRD into five major ethnic categories in line with the UK census groupings. These include: White (White British, White Irish, Any other White); Asian (Indian, Pakistani, Bangladeshi, Chinese, Other Asian); Black (Black Caribbean, Black African, Other Black); Mixed (White & Black Caribbean, White & Black African, White & Asian, Any other Mixed); and Other as ethnic group. For individuals with multiple ethnicity codes, we assigned a single best ethnicity based on the most frequently or most recently recorded codes [21,22]. Those with missing ethnicity records in both databases were excluded from the analyses.

Presenting features of possible underlying cancer: We identified features of possible cancer (Table 1) using codes in the CPRD [23], based on site-specific symptoms, signs, or blood-test results in the original or revised National Institute for Health and Care Excellence (NICE) guidance, NG12 (2015) [24,25]. We excluded features recorded more than one year before diagnosis, as these are less likely to relate to the cancer [26]. Each feature could be recorded in isolation (only ever recorded once) or recorded multiple times - contemporaneously (with other features) or recorded with other features at a different time. We included the first recorded incidence of each feature for each patient by site.

Other variables: The age, sex, and multi-morbidities of the patients were identified from the CPRD. Socioeconomic deprivation was defined using quintiles of the 2015 Index of Multiple Deprivation (IMD), a measure of relative deprivation for small areas in England [12,13]. Smoking status (current, smoker, or ex-smoker and unknown) was identified from the CPRD using codes for smoking status and smoking-cessation medication as previously reported [27]. Co-morbidities recorded before cancer diagnosis were identified from the CPRD, using medical codes relating to 36 long-term conditions described elsewhere [28]. Patients were categorised into five groups based on the general-outcome weighting Cambridge Multimorbidity Score (CMS) [29], with one group containing those with no included morbidities and the rest categorised according to quartiles of the CMS score.

### 2.2. Statistical Analysis

Mixed effects (“multilevel”) logistic regression models were fitted to examine ethnic differences in recorded features by cancer site—with the random intercept for practices to account for the clustering of patients within general practices. The primary analysis examined ethnic differences in each feature separately (one model per feature per cancer site) in the year before diagnosis, with adjustment for age, sex, comorbidity scores, IMD, and smoking status. Exploratory analyses were undertaken and stratified by whether features were isolated or seen in combination with other features. However, due to the limited sample sizes in ethnic minority groups, these analyses were not informative and are not reported here. All analyses were undertaken in Stata version 16.1 (StataCorp, College Station, TX, USA) and the reporting guided by the REporting of studies Conducted using Observational Routinely collected Data (RECORD) framework [30].

## 3. Results

### 3.1. Participant Characteristics

After applying exclusions (Appendix A), 130,944 patients with recorded features were included in the analysis. The characteristics and results of analyses regarding patients without recorded features are shown in Appendix A, Appendix A, Appendix A. Table 2 shows the demographics of patients with relevant features, 92% (120,885/130,944) of whom were White. Around sixty percent were males, ranging from 57% among Asians to 71% in the Black group. Asian and Black patients with cancer were, on average, younger, more likely to have never smoked, and to reside in more deprived neighbourhoods. The proportion with co-existing conditions was similar in the White, Black, Asian, and Mixed group but lower in the Other group (Table 2).

### 3.2. Ethnic Differences in the Number of Recorded Features

There was a total of 188,487 incident features in the year before diagnosis, around half (48%) were the only feature recorded for a patient (isolated features). The remainder were recorded in patients with one or more recorded features at some time in the year before diagnosis (multiple features). The number of recorded features differed considerably by site and ethnicity (Figure 1 and Appendix A). Breast cancer was dominated by a single feature (breast lump), while other sites had more diverse features, with patients often having multiple features. After adjustment, the odds of having multiple features were higher among Asian patients with breast [Adjusted Odds Ratio (AOR) = 1.56, 95% CI: 1.04–2.33], colorectal (AOR = 1.36, 95% CI: 1.11–1.67), and prostate cancer (AOR = 1.20, 95% CI: 1.02–1.40) compared with White patients (Figure 1). Similarly, Black patients with breast (AOR = 2.94, 95% CI: 1.94–4.46) and colorectal cancer (AOR = 1.48, 95% CI: 1.19–1.85) were more likely to have multiple features than White patients. The opposite was true of Black patients with lung cancer (AOR = 0.77, 95% CI: 0.62–0.95), and patients in the Other group with prostate cancer (AOR = 0.82, 95% CI: 0.71–0.95).

### 3.3. Ethnic Differences in the Type of Recorded Features by Site

Across all sites, the proportions with relevant recorded features were broadly similar by ethnicity, but with notable exceptions (Figure 2 and Appendix A). In breast cancer, there was weak evidence that Black patients were more likely than White patients to have breast pain [(AOR = 1.42, 95% CI: 0.98–2.07), Table 3]. For lung cancer, Black patients were less likely than White patients to have chest infections recorded (AOR = 0.74, 95% CI: 0.56–0.98). Asian patients were less likely to have dyspnoea (AOR = 0.78, 95% CI: 0.63–0.96) but were more likely to have cough (AOR = 1.33, 95% CI: 1.12–1.59) compared with White patients. Patients in the Mixed group were more likely than White patients to have dyspnoea (AOR = 1.23, 95% CI: 1.06–1.42). Those in the Other group were more likely to have thrombocytosis (AOR = 1.36, 95% CI: 1.19–1.56), but were less likely to have cough (AOR = 0.87, 95% CI: 0.76–0.98) and haemoptysis (AOR = 0.72, 95% CI: 0.54–0.96) compared with White patients.

For prostate cancer, Asian [AOR = 1.38, 95% CI: 1.02–1.88) and Black patients (AOR = 1.79, 95% CI: 1.46–2.19) were more likely than White patients to have erectile dysfunction. Black patients were less likely than White patients to have lower urinary tract infections (LUTS) (AOR = 0.85, 95% CI: 0.73–0.98) and haematuria (AOR = 0.65, 95% CI: 0.50–0.85). Patients in the Other ethnic group were more likely than White patients to have had a digital rectal examination (AOR = 1.62, 95% CI: 1.01–2.61) but were less likely to have erectile dysfunction (AOR = 0.59, 95% CI: 0.40–0.99).

For colorectal cancer, Asian (AOR = 0.51, 95% CI: 0.34–0.75), Black (AOR = 0.64, 95% CI: 0.43–0.95), and Mixed patients (AOR = 0.69, 95% CI: 0.49–0.97) were less likely to have changes in bowel habits than White patients. However, Asian patients were more likely to have iron deficiency (AOR = 1.27, 95% CI: 1.04–1.56), while patients in the Other ethnic group were more likely to have weight loss (AOR = 1.61, 95% CI: 1.15–2.25) compared to White patients.

For oesophagogastric cancer, Black (AOR = 1.34, 95% CI: 0.99–1.80) and Asian (AOR = 1.37, 95% CI: 0.99–1.89) patients were more likely to have low haemoglobin with gastrointestinal bleeding. Black (AOR = 1.39, 95% CI: 1.01–1.93) and Asian patients (AOR = 1.57, 95% CI: 1.14–2.17) were also more likely to have upper abdominal pain than White patients. Patients in the Other ethnic group were more likely to have thrombocytosis (AOR = 1.28, 95% CI: 0.99–1.65) and vomiting (AOR = 1.37, 95% CI: 0.99–1.87).

In myeloma, Black patients were more likely to have a record of an abnormal white blood cell count [AOR = 1.69, 95% CI: 1.22–2.36] but were less likely to have had back pain [AOR = 0.68, 95% CI: 0.48–0.96] than White patients. Asian patients were more likely to have abnormal erythrocyte sedimentation rates than White patients (AOR = 1.52, 95% CI: 1.02–2.25).

## 4. Discussion

This is the first UK study to examine ethnic differences in the profile of symptoms recorded before cancer diagnosis. The sample size was large and examined six common cancers. We used robust databases [12,13,14,15] and methods to identify variables included in our analyses. Information on patients’ ethnicities was identified from the CPRD and HES, with 99% completeness. However, we used combined ethnic categories in the analysis, recognising that this hides some differences across ethnic subgroups. More granular ethnicity categorisation would have reduced the power (particularly in rarer cancers) and made the interpretation of our findings unwieldy or impossible.

### Interpretation of Findings

Our finding that Asian and Black patients less frequently had relevant recorded features before diagnosis was unexpected, given that these patients use primary care more often [10,20]. This finding may indicate one or a combination of three possibilities: first, Asian and Black patients experience relevant NICE symptoms less often; second, Asian and Black patients may be less likely to disclose relevant symptoms during consultations, as previously shown in men with LUTS and erectile dysfunction [9]; third, GPs may be less likely to record features, or record features as free text rather than using codes, for these groups. Whichever the case may be, this finding raises important questions for future research to better understand the journeys (and associated outcomes) for Asian and Black patients with no apparent features (or non-NICE features) in primary care before diagnosis. Breast lump was the most common feature of breast cancer, consistent with previous studies [31,32,33], with little evidence of differences by ethnicity in the proportion with this feature. However, a small proportion of women had breast pain, which was recorded as an isolated feature in over half of Black women with that feature. These women may form the bulk of those experiencing greater pre-referral consultations and longer times to diagnosis [10,11], since isolated breast pain is less predictive of cancer [31,34,35], with no current UK recommendation for urgent investigation. Breast pain has been linked to a greater risk of advanced-stage breast cancer [33]. Therefore, while efforts to boost breast screening seem currently ineffective in Black women, a better understanding of the predictive value of breast symptoms in these women may help to improve their diagnostic experience.

As in previous studies [7,32,36,37], lung cancer was characterised by multiple features, with inconsistent evidence of ethnic differences.

As previously reported [32,36], raised prostate-specific antigen (PSA) was the most common feature of prostate cancer—mostly recorded as an isolated feature and with little evidence of differences by ethnicity. This finding is notable, considering public interest in PSA screening for men at risk of prostate cancer, especially Black men. The impact of such an intervention may be limited, if opportunistic screening with PSA tests is common, as our finding here suggests. An estimated 65% of all first PSA tests in the UK are due to opportunistic screening; [38] this is comparable to the proportion with isolated raised PSAs in our study (69%). Notably, Asian and Black men were more likely than White men to have erectile dysfunction, contrary to a previous small study showing that men from the former groups may not disclose erectile dysfunction during GP consultations [9]. Given that the majority of men with these features had them synchronously with other features, thereby simplifying GPs’ investigation decisions, future research to improve prostate-cancer outcomes in Black men could focus more on ways to maximise the usage of PSA testing in primary care. However, LUTS have been linked to early-stage prostate cancer [33]; therefore, improving awareness of these features may help improve outcomes in Black men.

Our finding of the multiple features in colorectal cancer aligns with earlier reports [36,37,39], but our study is the first to report ethnic differences in colorectal-cancer features. We found that patients of Asian, Black, and Mixed ethnic groups were less likely than White patients to have a record of changes in bowel habits, although the Asian group were more likely to have iron-deficiency anaemia. Both features have low PPVs for cancer in the general population, except when accompanied by other features [40]. In this study, over half of Asian and Black patients with changes in bowel habits and two-thirds of Asians with iron-deficiency anaemia had these recorded as isolated features, which may explain the recent report of longer times to diagnosis in these groups compared with the White group [11].

Oesophagogastric cancer was also characterised by multiple features, similar to those found in previous studies [36,37,41]. Black and Asian patients were more likely than White patients to have upper abdominal pain and low haemoglobin. Both abdominal pain and low haemoglobin have low PPVs for cancer, with a recommendation of non-urgent access to specialist investigation [22]. Again, these findings may, in part, explain the longer times to diagnosis in Asian and Black patients compared to White patients [11].

Myeloma was also characterised by multiple features, similar to those in an earlier report [36,37,42]. We found little evidence of a difference by ethnicity, except for back pain and abnormal white blood cell counts. An abnormal white blood cell count has a low PPV when reported as an isolated feature [42], although it was recorded synchronously with other features in over half of the Black patients in our cohort.

We urge cautious interpretation of our findings regarding the Other ethnic group, given the differences within the Other ethnic group, with no prior UK studies specifically exploring cancer inequalities in this group.

Overall, these ethnic differences in recorded features may reflect variations in cancer biology, particularly for breast and prostate cancer, for which there is evidence of ethnic differences [43,44]. They may also be related to patient factors such as symptoms awareness, fear of cancer, stigma, religion or belief, health literacy, emotional and language barriers, and socioeconomic deprivation; all these factors are associated with cancer symptoms appraisal and medical help-seeking in ethnic minority groups [45,46,47,48,49,50]. Alternatively, the findings may reflect healthcare system factors, including GP preferences for coding symptoms, GP attitudes toward patients, appointment-scheduling problems, the short duration of consultations, and continuity of care, all of which may be associated with symptoms disclosure or recording during consultation [9,48,51]. The extent to which (each or a combination of) these factors contribute to the observed ethnic differences in recorded features is uncertain.

## 5. Conclusions

We found small, but potentially important, evidence of ethnic differences in cancer features before cancer diagnosis. For some sites, patients with Asian and Black ethnic backgrounds were more likely to have low-risk features than White patients. Indeed, there was no site where either group was more likely than the White group to have alarm features. These findings may explain, at least in part, ethnic differences in cancer diagnostics, and stresses the need for further exploration of the predictive value of cancer features in primary care by ethnicity—alongside the associated impacts on cancer outcomes. Existing interventions, such as awareness campaigns (including non-alarm symptoms), may help to improve the reporting of symptoms in Asian and Black groups, especially if targeted at these communities. A revision of investigation recommendations may be necessary to ensure that they reflect ethnic variations in cancer risks, thereby fostering equity in specialist referral.

## Figures and Tables

**Figure 1 cancers-15-03100-f001:**
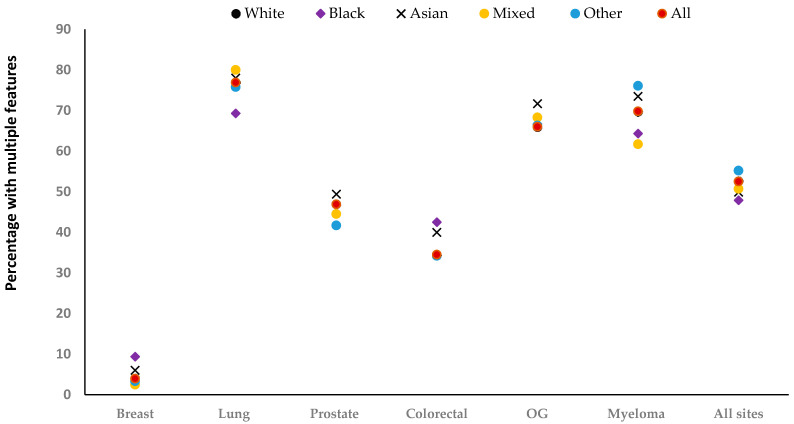
Percentage with multiple relevant features recorded, by ethnicity. OG: Oesophagogastric. The total number of recorded features for each site: [breast (*n* = 21,305), lung (*n* = 61,192), prostate (*n* = 52,039), colorectal (*n* = 29,033), oesophagogastric (*n* = 19,136), myeloma (*n* = 5782), and All sites (*n* = 188,487].

**Figure 2 cancers-15-03100-f002:**
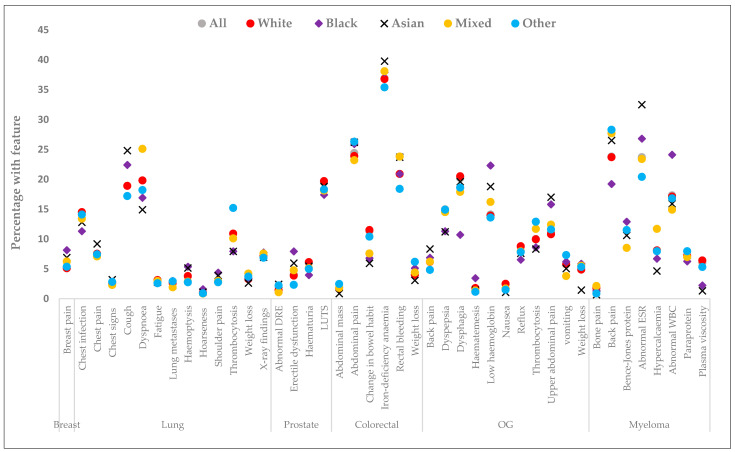
The most common relevant features recorded at least once - up to 12 months before diagnosis, by ethnicity. OG: Oesophagogastric; LUTS: lower urinary tract symptoms; WBC: white blood cell count; ESR: erythrocyte sedimentation rates; DRE: digital rectal examination. Total number of recorded features each site: [breast (*n* = 21,305), lung (*n* = 61,192), prostate (*n* = 52,039), colorectal (*n* = 29,033), oesophagogastric (*n* = 19,136), myeloma (*n* = 5782), and All sites (*n* = 188,487]. Breast lump was recorded in 91% and raised PSA value in 69% of patients, with no evidence of ethnic differences. Actual percentages are included in Appendix A.

**Table 1 cancers-15-03100-t001:** Cancer features sought in participants’ medical records in the year before diagnosis.

Cancer Site	NICE Features
**Breast**	Breast pain, breast lump, breast skin changes (peau d’orange), nipple discharge, nipple retraction, lymphadenopathy (axilla).
**Lung**	Appetite loss, chest infection, chest pain, chest signs consistent with lung cancer, cough, dyspnoea, fatigue, features suggestive of lung metastases’ finger clubbing, haemoptysis, hoarseness, lymphadenopathy (supraclavicular, cervical), shoulder pain, signs of superior vena cava obstruction, stridor, thrombocytosis, weight loss, x-ray findings suggestive of lung cancer.
**Prostate**	Abnormal digital rectal examination, erectile dysfunction, haematuria (visible), nocturia, raised prostate specific antigen (PSA) above age-specific value, lower urinary tract symptoms (LUTS)—frequency, urinary hesitancy, urinary retention, urinary urgency.
**Colorectal**	Abdominal mass, abdominal pain, change in bowel habit, faecal occult blood, iron-deficiency anaemia, rectal bleeding, rectal mass, weight loss.
**Oesophagogastric**	Back pain, dyspepsia, dysphagia, haematemesis, gastrointestinal bleeding, low haemoglobin, nausea, reflux, suspicious barium meal results, thrombocytosis, upper abdominal mass, upper abdominal pain, vomiting, weight loss.
**Myeloma**	Bone pain, back pain, Bence-jones protein, abnormal erythrocyte sedimentation rate (ESR), hypercalcaemia, abnormal white cell count, pathological fracture, plasma viscosity consistent with myeloma, protein electrophoresis suggesting myeloma, spinal cord compression suspected of being caused by myeloma.

**Table 2 cancers-15-03100-t002:** Participants characteristics.

		White	Black	Asian	Mixed	Other	All
Age (years)	Median (IQR)	73 (64–80)	68 (56–76)	67 (56–75)	71 (63–79)	73 (63–81)	72 (64–80)
Sex	Male *n* (%)	72,678 (60.1)	1812 (71.1)	1193 (56.7)	1491 (60.6)	1707 (57.4)	78,881 (60.2)
IMD *n* (%) *	*1 (least deprived)*	29,625 (24.5)	124 (4.87)	332 (15.8)	462 (18.8)	732 (24.6)	31,275 (23.9)
	*2*	27,192 (22.5)	167 (6.56)	353 (16.8)	504 (20.5)	699 (23.5)	28,915 (22.1)
	*3*	24,087 (19.9)	429 (16.8)	453 (21.5)	462 (18.8)	589 (19.8)	26,020 (19.9)
	*4*	20,693 (17.1)	724 (28.4)	445 (21.1)	512 (20.8)	511 (17.2)	22,885 (17.5)
	*5 (most deprived)*	19,211 (15.9)	1103 (43.3)	522 (24.8)	520 (21.1)	443 (14.9)	21,799 (16.7)
Morbidity score *n* (%)	*0—None*	8203 (6.79)	162 (6.36)	132 (6.27)	129 (5.24)	384 (12.9)	9010 (6.88)
	*1*	23,043 (19.1)	531 (20.8)	390 (18.5)	374 (15.2)	749 (25.2)	25,087 (19.2)
	*2*	23,750 (19.7)	603 (23.7)	483 (22.9)	459 (18.7)	649 (21.8)	25,944 (19.8)
	*3*	29,956 (24.8)	620 (24.3)	537 (25.5)	646 (26.3)	679 (22.8)	32,438 (24.8)
	*4 (highest burden)*	35,903 (29.7)	632 (24.8)	563 (26.8)	852 (34.6)	515 (17.3)	38,465 (29.4)
Smoking status, *n* (%)	*Current smoker*	22,154 (18.3)	498 (19.5)	261 (12.4)	467 (18.9)	732 (24.6)	24,112 (18.4)
	*Never smoked*	41,684 (34.5)	1293 (50.8)	1324 (62.9)	829 (33.7)	1020 (34.3)	46,150 (35.2)
	*Ex-smoker*	44,830 (37.1)	549 (21.6)	337 (16.0)	841 (34.2)	951 (31.9)	47,508 (36.3)
	*Unknown*	12,187 (10.1)	208 (8.16)	183 (8.69)	323 (13.1)	273 (9.17)	13,174 (10.1)
Sites, *n* (%)	*Breast*	19,113 (15.8)	387 (15.2)	548 (26.0)	443 (18.0)	385 (12.9)	20,876 (15.9)
	*Lung*	29,899 (24.7)	289 (11.3)	368 (17.5)	524 (21.3)	961 (32.3)	32,041 (24.5)
	*Prostate*	35,938 (29.7)	1237 (48.6)	582 (27.7)	853 (34.7)	673 (22.6)	39,283 (30.0)
	*Colorectal*	22,128 (18.3)	320 (12.6)	358 (17.0)	411 (16.7)	530 (17.8)	23,747 (18.1)
	*Oesophagogastric*	10,728 (8.88)	175 (6.87)	165 (7.84)	168 (6.83)	366 (12.3)	11,602 (8.86)
	*Myeloma*	3049 (2.52)	140 (5.49)	84 (3.99)	61 (2.48)	61 (2.05)	3395 (2.59)
	**Total**	**120,885 (92.3)**	**2548 (1.95)**	**2105 (1.61)**	**2460 (1.88)**	**2976 (2.27)**	**130,944 (100)**

IMD: Index of Multiple Deprivation; * Missing record of IMD [*n* = 50 (0.04%)].

**Table 3 cancers-15-03100-t003:** Crude and adjusted odds ratio for the association between recorded features up to 12 months before diagnosis and ethnicity.

Sites	Features	Black				Asian				Mixed				Other			
		OR	AOR	95% CI	*p*-Value	OR	AOR	95% CI	*p*-Value	OR	AOR	95% CI	*p*-Value	OR	AOR	95% CI	*p*-Value
Breast	Breast pain	1.65	1.42	0.98–2.07	0.06	1.38	1.14	0.82–1.60	0.43	1.24	1.19	0.81–1.77	0.36	1.05	1.11	0.71–1.73	0.65
	Breast lump	0.72	0.76	0.55–1.06	0.11	0.95	1.07	0.79–1.46	0.66	0.93	0.94	0.67–1.31	0.71	0.97	0.93	0.65–1.34	0.70
Lung	Chest infection	0.76	0.74	0.56–0.98	0.04	0.88	0.86	0.69–1.08	0.19	0.89	0.89	0.74–1.06	0.19	0.99	1.01	0.88–1.16	0.89
	Chest pain	1.01	0.90	0.64–1.27	0.55	1.25	1.15	0.89–1.49	0.29	0.94	0.95	0.74–1.20	0.65	0.98	1.01	0.84–1.21	0.93
	Chest signs	1.15	1.09	0.63–1.89	0.75	1.32	1.07	0.69–1.64	0.77	0.95	1.05	0.69–1.59	0.80	1.12	1.03	0.77–1.38	0.84
	Cough	1.27	1.23	0.98–1.53	0.07	1.39	1.33	1.12–1.59	0.002	0.92	0.96	0.82–1.14	0.65	0.89	0.87	0.76–0.98	0.03
	Dyspnoea	0.82	0.98	0.77–1.25	0.87	0.71	0.78	0.63–0.96	0.02	1.36	1.23	1.06–1.42	0.005	0.89	1.02	0.89–1.15	0.77
	Fatigue	0.99	1.09	0.65–1.85	0.73	0.89	0.93	0.59–1.46	0.75	0.96	0.97	0.68–1.39	0.88	0.82	0.79	0.58–1.06	0.11
	Lung metastases	1.14	1.08	0.64–1.81	0.79	0.84	0.83	0.50–1.38	0.48	0.72	0.73	0.46–1.13	0.16	1.12	1.09	0.82–1.45	0.54
	Haemoptysis	1.43	1.33	0.89–1.98	0.17	1.37	1.34	0.96–1.88	0.09	0.75	0.77	0.54–1.12	0.17	0.72	0.72	0.54–0.96	0.02
	Shoulder pain	1.52	1.44	0.93–2.23	0.10	1.34	1.24	0.85–1.83	0.27	1.02	1.04	0.72–1.49	0.85	0.95	0.96	0.72–1.29	0.79
	Thrombocytosis	0.72	0.74	0.53–1.03	0.08	0.72	0.78	0.59–1.03	0.08	0.91	0.94	0.77–1.16	0.58	1.47	1.36	1.19–1.56	<0.0001
	Weight loss	1.03	0.99	0.60–1.62	0.96	0.79	0.81	0.51–1.29	0.37	1.29	1.25	0.92–1.71	0.15	1.15	1.07	0.83–1.39	0.58
	X-ray findings	0.97	0.89	0.63–1.29	0.56	0.96	0.94	0.68–1.28	0.68	0.95	1.01	0.79–1.28	0.96	0.98	0.93	0.76–1.13	0.44
Prostate	Abnormal DRE	0.82	0.81	0.49–1.34	0.42	1.54	1.56	0.95–2.56	0.08	0.78	0.78	0.43–1.40	0.41	1.62	1.62	1.01–2.61	0.05
	ED	2.03	1.79	1.46–2.19	<0.0001	1.54	1.38	1.02–1.88	0.04	1.23	1.13	0.85–1.50	0.39	0.59	0.63	0.40–0.99	0.04
	Haematuria	0.65	0.65	0.50–0.85	0.002	0.90	0.89	0.65–1.23	0.50	0.83	0.82	0.63–1.08	0.16	0.81	0.86	0.63–1.18	0.36
	LUTS	0.86	0.85	0.73–0.98	0.02	0.98	0.97	0.80–1.17	0.75	0.93	0.93	0.79–1.09	0.38	0.93	0.93	0.78–1.11	0.44
	Raised PSA	1.05	1.08	0.95–1.22	0.23	0.92	0.94	0.79–1.09	0.42	1.08	1.09	0.96–1.25	0.18	1.16	1.12	0.96–1.31	0.15
Colorectal	Abdominal pain	1.12	1.04	0.83–1.31	0.71	1.13	1.03	0.83–1.28	0.80	0.96	0.93	0.75–1.16	0.53	1.13	1.07	0.89–1.28	0.46
	CBH	0.57	0.64	0.43–0.95	0.03	0.49	0.51	0.34–0.75	0.001	0.64	0.69	0.49–0.97	0.03	0.90	0.83	0.64–1.08	0.17
	Iron-deficiency	0.99	1.06	0.85–1.33	0.58	1.10	1.27	1.04–1.56	0.02	1.04	1.03	0.85–1.25	0.77	0.90	0.99	0.84–1.18	0.96
	Rectal bleeding	1.00	0.92	0.72–1.18	0.51	1.17	1.04	0.83–1.30	0.72	1.17	1.16	0.94–1.44	0.16	0.86	0.85	0.69–1.04	0.12
	Weight loss	1.26	1.27	0.80–2.02	0.31	0.75	0.82	0.47–1.41	0.47	1.08	1.07	0.69–1.66	0.77	1.61	1.61	1.15–2.25	0.005
OG	Back pain	1.12	1.09	0.69–1.74	0.69	1.38	1.31	0.85–2.03	0.22	1.01	0.95	0.59–1.54	0.83	0.77	0.85	0.58–1.24	0.39
	Dyspepsia	0.74	0.78	0.54–1.13	0.19	0.73	0.70	0.48–1.03	0.07	0.97	1.01	0.72–1.40	0.97	1.01	0.99	0.79–1.25	0.94
	Dysphagia	0.46	0.73	0.49–1.08	0.11	0.95	1.19	0.87–1.63	0.29	0.85	0.91	0.67–1.25	0.57	0.88	0.86	0.69–1.07	0.16
	Low haemoglobin	1.78	1.34	0.99–1.80	0.05	1.43	1.37	0.99–1.89	0.05	1.20	1.16	0.84–1.61	0.37	0.98	1.08	0.84–1.39	0.55
	Reflux	0.73	0.80	0.49–1.29	0.36	0.86	0.84	0.54–1.32	0.46	0.89	0.94	0.61–1.44	0.76	0.88	0.87	0.64–1.18	0.36
	Thrombocytosis	0.85	0.74	0.48–1.13	0.16	0.82	0.79	0.51–1.23	0.30	1.22	1.17	0.81–1.69	0.40	1.35	1.28	0.99–1.65	0.05
	Upper abdominal pain	1.55	1.39	1.01–1.93	0.04	1.69	1.57	1.14–2.17	0.006	1.17	1.15	0.80–1.64	0.45	1.09	1.05	0.81–1.36	0.7
	Vomiting	1.08	0.93	0.57–1.51	0.76	0.87	0.82	0.47–1.41	0.47	0.64	0.62	0.34–1.15	0.13	1.29	1.37	0.99–1.87	0.05
Myeloma	Back pain	0.76	0.68	0.48–0.96	0.03	1.15	1.03	0.71–1.49	0.89	1.23	1.15	0.72–1.82	0.55	1.27	1.35	0.89–2.04	0.17
	Bence-Jones protein	1.07	1.11	0.68–1.80	0.69	0.97	1.03	0.57–1.87	0.92	0.86	0.86	0.39–1.89	0.71	1.01	1.00	0.53–1.90	0.99
	Abnormal ESR	1.10	1.12	0.79–1.59	0.54	1.47	1.52	1.02–2.25	0.04	0.94	0.96	0.57–1.61	0.87	0.84	0.82	0.49–1.34	0.42
	Hypercalcaemia	0.82	0.78	0.45–1.34	0.37	0.55	0.57	0.27–1.24	0.16	1.50	1.49	0.78–2.83	0.22	0.98	0.99	0.49–1.98	0.98
	Abnormal WBC	1.54	1.69	1.22–2.36	0.002	0.92	0.97	0.62–1.53	0.90	0.85	0.89	0.49–1.58	0.69	0.98	0.95	0.57–1.56	0.82
	Paraprotein	0.89	0.99	0.55–1.77	0.96	1.07	1.13	0.59–2.18	0.71	0.90	1.01	0.43–2.38	0.99	1.11	1.08	0.53–2.19	0.84

LUTS: lower urinary tract symptoms, ED: erectile dysfunction, PSA: prostate-specific antigen, WBC: white blood cell count, ESR: erythrocyte sedimentation rates; CBH: Change in Bowel Habit, DRE: digital rectal examination, OG: Oesophagogastric. The White group is the reference ethnic group; OR—crude odds ratio; AOR—adjusted odds ratio. Adjusted for age, sex, IMD, smoking status and comorbidity.

## Data Availability

This study used CPRD-linked data, access to which is subject to protocol approval by an Independent Scientific Advisory Committee, and under which conditions data cannot be shared directly.

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
