# Peer review of "Are There Ethnic Differences in Recorded Features among Patients Subsequently Diagnosed with Cancer? An English Longitudinal Data-Linked Study"

_cancers, 2023, doi:10.3390/cancers15123100_

Round 1

Reviewer 1 Report

Martins et al., have highlighted ethnic differences in presenting features recorded in primary care before cancer diagnosis.

Overall the article is structured and is communicable to the specific audience. However, there are a number of minor issues that need to be addressed to enhance the quality of the article.

1.      The authors should provide a simple summary of their research perspectives in one paragraph before the Abstract in layman’s terms.

2.      The authors should provide three to ten keywords.

3.      The authors may reconsider putting their designations in the author section.

4.      The authors are suggested to improve the picture quality in Figure 1 and Figure 2. The figure labels should be clearly visible.

5.      The authors are suggested to lower the font size in Table 2 to accommodate the values in one row.

6.      The authors need extensive formatting in references. The authors should keep same reference style for every reference. The authors should remove the hyperlinks from the references.

Author Response

  • The authors should provide a simple summary of their research perspectives in one paragraph before the Abstract in layman’s terms. Response: The following Simple summary now included

“This UK population-based study aimed to determine whether presenting features of cancer recorded in primary care before diagnosis differed by ethnicity. We found that for some cancer types, Asian and Black patients were more likely than White patients to have ‘less concerning’ features like cough and upper abdominal pain recorded before diagnosis. Indeed, there was no site where either group was more likely than the White group to have alarm features like blood in urine recorded. However, further research is necessary to determine the extent to which these ethnic differences reflect the disease biology, patient, or healthcare factors“

  • The authors should provide three to ten keywords. Response: we have added six keywords to the manuscript

“ethnic inequalities; cancer symptoms; cancer diagnosis; primary care; diagnostic pathway; symptomatic cancer”

  •  The authors may reconsider putting their designations in the author section. Response: authors designations now removed 
  •  The authors are suggested to improve the picture quality in Figure 1 and Figure 2. The figure labels should be clearly visible. Response: Figures 1&2 replaces with editable versions. 
  •  
  • The authors are suggested to lower the font size in Table 2 to accommodate the values in one row. Response: Font size reduced as advised.
  •  The authors need extensive formatting in references. The authors should keep same reference style for every reference. The authors should remove the hyperlinks from the references. Response: References now revised, with the hyperlinks removed.

Reviewer 2 Report

The manuscript goes to several important areas for primary prevention. That is the importance of the primary physician (aka GP) and accurate revelations from patients in terms of symptoms. Also, important as well is features prognostic of certain cancers may lead to better detection strategies. The authors conduct a mixed effect logistic regression model on English cancer registry to look at aggregate groups (White, Asian, Black, mixed, other) for six common cancers (breast, prostate, lung, colorectal, esophagogastric, myeloma) to investigate if there were differences in features between group up to one year prior to diagnosis.

Originality/Novelty: The study is novel in that takes a deep dive with a large sample size; though as authors note primary, GP, physicians rely on instinct and patient report as well.

Significance: Moderately significance. Primary prevention through better detections strategies is significant to cancer prevention and control. Added to this is patient self-report. These are areas worthy of future interest. .

Quality of Presentation: The overall quality of the presentation is excellent.  

Scientific Soundness: Science is a little difficult to generalize or make conclusions of, but the analysis and results reported are based on good statistical technique..

Interest to the Readers: I think this article will be of great interest to most readers of this journal.

Overall Merit: There is merit to publishing the work as the authors found small, potentially important evidence and could lead to future studies to further differentiate strategies for detection for certain groups or heighten physician attention to possibilities of follow-up.  

English Level: Well written – excellent ‘flow’

The only suggestions I would have is in the materials and methods section. Suggest more explanation on what is the actual scope and what are the strengths of the CPRD-linked data as I am not familiar with it (instead of statement “well documented”)

Similarly, the English breakdown of ethnic groups elaborated a little more. As well as a few sentences on IMD (deprivation measure) would be helpful for those like me unfamiliar with these measures.  

Author Response

We thank the reviewer for the positive feedback.

  • The only suggestions I would have is in the materials and methods section. Suggest more explanation on what is the actual scope and what are the strengths of the CPRD-linked data as I am not familiar with it (instead of statement “well documented”). Response: We have now expanded the methods section with the following statements: “The CPRD-Aurum comprises routinely gathered data from 890 consenting English practices (August 2019 release), with over 28 million patients eligible for linkage to other health care databases.12 13  It includes coded and anonymised data on patients’ medical history, including symptoms, investigations, diagnoses, prescriptions, referral, and demographics (e.g. age, gender, and ethnicity).12 13 HES data - Admitted patient care and Outpatient elective - contains medical records on all hospital admissions and outpatient appointments in England. 12-14 The NCRAS cancer registry data includes records of all tumours diagnosed in England, alongside information about treatment and patient-reported outcomes. 15”
  • Similarly, the English breakdown of ethnic groups elaborated a little more. As well as a few sentences on IMD (deprivation measure) would be helpful for those like me unfamiliar with these measures. Response: We have expanded the statements regarding ethnicity and IMD with the following texts: “Briefly, we extracted and collapsed all ethnicity records from the CPRD into five major ethnic categories in line with the UK census groupings. These includes: White (White British, White Irish, Any other White); Asian (Indian, Pakistani, Bangladeshi, Chinese, Other Asian); Black (Black Caribbean, Black African, Other Black); Mixed (White & Black Caribbean, White & Black African, White & Asian, Any other Mixed); and Other ethnic group. For individuals with multiple ethnicity codes, we assigned a single best ethnicity based on the most frequently or most recently recorded codes.21 22 Those with missing ethnicity records in both databases were excluded from analyses.”

“Socioeconomic deprivation was defined using quintiles of the 2015 Index of Multiple Deprivation (IMD), a measure of relative deprivation for small areas in England.12-13  “

Reviewer 3 Report

The article contains a few minor mistakes, for example: of a primary care consultations.
The singular verb was does not appear to agree with the plural subject Differences. Analysis and Diagnosis have singular or plural forms.

Revise the article thoroughly and remove minor mistakes, please.

Author Response

The article contains a few minor mistakes, for example: of a primary care consultation. The singular verb was does not appear to agree with the plural subject Differences. Analysis and Diagnosis have singular or plural forms.

Response: Manuscript has now edited, thank you.